# "Am I representative (of my age)? No, I'm not"—Attitudes to technologies and technology development differ but unite individuals across rather than within generations

**Sofi Fristedt** [1,2]☺*, **Samantha Svärdh**[3]☺, **Charlotte Löfqvist**[1]☺, **Steven M. Schmidt**[1]‡, **Susanne Iwarsson**[1]‡

1 Department of Health Sciences, Faculty of Medicine, Lund University, Lund, Sweden, 2 Jönköping Academy for Improvement of Health and Welfare, School of Health and Welfare, Jönköping University, Jönköping, Sweden, 3 City of Malmö, Malmö

☺ These authors contributed equally to this work.
‡ These authors also contributed equally to this work.
* sofi.fristedt@med.lu.se

**Data Availability Statement:** Data availability The data used in this study contains sensitive information about the study participants and they

## Abstract

While a broad spectrum of technologies is integrated in everyday life and routines, most research on ageing, health and technology has focused on attitudes toward and adoption of digital technologies including e-health, or home based monitoring systems. The aim of this study was to explore differences and similarities in attitudes and experiences with different types of technologies and development within and between three generations. We applied a qualitative, descriptive design and recruited a purposeful sample of participants from three generations (30–39, 50–59, 70–79 year old). The 25 participants took part in 3 x 2 focus groups. Forming four categories, the findings show that technologies enable as well as complicate everyday life. Participants expressed trust as well as uncertainty about risks when using technology and stated that use of digital services is required while support is limited. They identified that technology development is inevitable but not always in the service of users. In conclusion, experiences of and attitudes towards technologies and technology development are not limited to generation; perspectives sometimes unite individuals across rather than within generations. Thus future technologies and technology development, as well as services and policies aiming to support the use of said technologies should consider individual user perspectives including needs, desires, beliefs or goals neglected in the existing technology models, and involve users beyond generations defined by chronological age. Such strategies are likely to be more successful in supporting development of technologies usable for all.

did not provide consent for public data sharing. The current approval by the Regional Ethical Board in Lund, Sweden Dnr 2018/456 does not include data sharing. A minimal data set could be shared by request from a qualified academic investigator for the sole purpose of replicating the present study, provided the data transfer is in agreement with EU legislation on the general data protection regulation and approval by the Swedish Ethical Review Authority. Contact information: Department of Health Sciences, Lund University Box 157, 221 00 Lund, Sweden Att. Christina Brogårdh, Head of Department: christina.brogardh@med.lu.se aria_H.Nilsson@med.lu.se Principal investigator: Professor Susanne Iwarsson susanne.iwarsson@med.lu.se Swedish Ethical Review Authority, Box 2110, 75 002 Uppsala, Sweden. Phone: +46 10 475 08 00.

**Funding:** The present study is part of the GenerationTech project, funded by the Swedish Research Council for Health, Working Life and Welfare (FORTE) (contract no. 2017-01614). In addition, professor Susanne Iwarsson was financed by the Ribbingska Foundation in Lund.

**Competing interests:** The authors have declared that no competing interests exist.

## Introduction

Technology is omnipresent in everyday life for people of all ages, from the smart phones in our pockets to the refrigerators in our kitchens [1]. Population ageing across the globe has made adaptation and use of technologies a pertinent issue to enhance active and healthy ageing, and support health and social care services [2–4]. While technologies are broadly defined as goods, services, knowledge, skills etc. [5], the present study focuses on goods and related services.

A broad spectrum of technologies is well integrated in people's everyday life and routines, but most research on ageing, health and technology has focused on attitudes toward and adoption of digital technologies such as information and communication technology (ICT), e-health, wearable monitoring devices, home based monitoring systems, or smart home technology [6].

Previous research scarcely considered attitudes and use of technologies within and across age cohorts [7, 8], and research applying a generational perspective on a broad range of technologies is non-existent. Our study departed from Lim's [9] conclusion that generations are formed by technology used by people belonging to certain age cohorts during their formative period (10–25 years of age), and belonging to different technological generations explains why older adults find present day devices difficult to use. We acknowledge that definitions of generations may are changeable, and a process-oriented *"redefinition of and by generations in the course of time"* [10, p. 5] is influenced by technology development and media.

Younger generations have been defined as "digital natives" born and/or raised in already digitized environments, whereas "digital immigrants" denotes adults with experience from the analogue to digital transition [11]. Although older adults largely perceive digital technology positively, they utilize digital technology at lower rates when compared to younger generations and report less confidence, interest and usability [12]. In Sweden, 69% of citizens older than 75 years use the Internet, while usage is almost universal (93%) among those aged 65–75 years [13]. Household income, rural-living and lower levels of education are other factors limiting technology adoption and use [13, 14]. In addition, products developed specifically for use in later life may bring stigmatizing symbolism potentially preventing user adoption [8, 15]. Attitudes toward technology among different populations is therefore a complex phenomenon deserving better understanding, and several current studies [16, 17] largely focus on various health outcomes with a particular focus on older rather than ageing populations.

Previous studies on development of technologies [9] and attitudes toward technology use have largely utilised deductive lenses [2] such as the Technology Acceptance Model (TAM) [18, 19] or the Unified Theory of Acceptance and Use of Technology (UTAUT) [20]. In addition, most of these originate from disciplines such as information system engineering, business or management science. They neglect or only implicitly consider desires, beliefs and goals of potential users [21]. In contrast but in line with the underpinnings of gerontechnology, inter-disciplinary efforts integrating multiple disciplinary and user perspectives are certainly called for to address this neglect [14]. In fact, few studies have adopted qualitative, user-centred research designed to create a deeper understanding of attitudes toward technologies, especially in the light of differences and similarities between and across age cohorts or generations. Research considering ageing adults' use, perceptions and acceptance of a broad range of technologies applying a generational perspective is limited and called for [1, 8, 9]. Knowledge about the attitudes of people from different generations is necessary to inform the design of current and future technologies and enhance user experiences [22, 23]. The aim of this study was to explore differences and similarities in attitudes and experiences with different types of technologies and development within and between three generations.

## Method

This study was the first step of the GenerationTech project and served as the starting point for a series of studies on attitudes to technology from a generational perspective. We applied a qualitative descriptive design based on focus groups to generate new knowledge, useful in its own right as well as to inform a future questionnaire for a population-based survey and hypotheses in future research. Focus groups were chosen to capture a wide range of attitudes and experiences through the dynamics of group discussions than would have been possible through individual interviews [24]. Designing a process with age homogenous groups in the first round of sessions and age-mixed groups in the second, focus groups enabled discussions within and across generations. In this way, we were able to achieve engaging and thoughtful discussions among the participants and thereby a deeper understanding of the topics at target. Ethical approval for the present study was granted by the Regional Ethical Board in Lund **Dnr 2018/456**.

### Participants

Addressing three generations (30–39, 50–59 and 70–79 years old) we recruited a purposeful sample of participants able to communicate in Swedish. Aiming for heterogeneity [24] in attitudes and experiences with technologies from a broad perspective, we recruited the participants from different work and life situations; through networks such as social service staff in a nearby municipality as well as non-governmental organizations for people with disabilities. With support from university staff, we recruited current/past students within humanities, social medicine, global health and engineering. We also approached professionals and seniors who had previously attended events organized by our research centre; we knew that they had experience from welfare technology as well as assistive technology. Homogeneity [24] in terms of age and heterogeneity [24] based on sex and educational background, work and life situations was established to generate and explore different perspectives on the study topic [25].

All potential participants received an informational invitation e-mail. Those who expressed interest received a detailed information letter, which included the time for the first focus group session. Those who were interested and able to attend on the specified date were phoned to establish personal contact. Additional information about the study was then provided, and verbal consent to participate in the study was established. At the start of the first focus group session, all participants signed written informed consent.

The final sample included 25 participants: nine aged 30–39 years (4 men, 5 women; mean age 36.0), six aged 50–59 years (1 man, 5 women; mean age 54.5) and ten aged 70–79 years (5 men, 5 women; mean age 75.5). Nineteen participants had a university degree, and the other six had at least a high school education. Ten lived alone and the rest were cohabiting. A majority of the participants reported b in good or very good health. Together they represented a diversity of professions.

### Data collection

We developed a set of open-ended questions [24] and asked two men and two women (aged 70–79) from the User Board at our research centre for feedback on the clarity and interpretation of the questions. This led us to a slight change of terminology and the addition of a short introduction regarding everyday technologies. The final questioning route focused on attitudes and lifetime experiences of technology use and developments based on the following key questions:

- What types of technologies are most important for you?

- What types of technologies and developments have had the greatest impact on you in a long-term perspective?

- What influences your choice of technologies?

- What enables or complicates the use of technologies?

- What are your thoughts on the rapid technology development seen today?

The first round of focus groups included three age-homogenous groups (i.e., 30–39, 50–59 and 70–79 year olds) with 6–10 people in each. Each interview lasted two hours and was moderated by CL (third author), with SS (second author) acting as assistant moderator, and SF (first author) as co-assistant moderator during one of the sessions.

A second round of focus groups was part of the original plan to allow member checking, confirmation and elaboration of findings from the first round. Upon preliminary analysis of and reflection on the first round we recognized the need to compare and contrast generational perspectives in a more dynamic way and therefore chose to create heterogeneous groups based on age and sex to support this [24]. A second set of open-ended questions was developed:

- Beyond user-friendliness, what defines simple technologies, and facilitates/complicates use?

- What influences/motivates the decision to use certain products?

- What are your current and future expectations or hopes for technology?

- What are your thoughts about integrity when using digital technologies?

- Do we have to keep up with the technological development? If so, who should ensure that?

- How to navigate the technological development? Support needed?

- What future needs of technologies do you expect?

We contacted all participants again via email to confirm or decline further participation. Seventeen, i.e., seven men and ten women representing 30–39 year olds (1 man, 2 women) 50–59 year olds (1 man, 4 women) and 70–79 year olds (4 men, 5 women) agreed and had the possibility to participate while eight were not able/willing to attend due to other engagements, impairing health issues, or lack of response to inquiries. We mixed the 17 participants into three groups, each representing the three generations and both sexes, also considering their availability on specific dates and times. However, as a man aged 30–39 made a late cancellation, one of the focus groups included participants solely from the two older generations. During the second round, SS acted as the moderator and CL as assistant moderator for two of the sessions, and SF for the third.

During all focus groups, we asked additional questions to stimulate discussions or clarify for more depth and gave the participants free rein to discuss organically. We offered complimentary refreshments and travel reimbursements. All focus groups were digitally recorded.

## Data analysis

We transcribed the recordings verbatim and entered them into the NVivo software to support a qualitative content analysis [26]. Qualitative content analysis is a flexible method appropriate when analysing data from focus groups [22] particularly relevant related to these multifaceted phenomena of technology where little knowledge exists from the generational perspective. As condensing meaning units is redundant when using NVivo, the analysis procedure was slightly adapted. Initially, SS read all data several times, identified meaning units and assigned similar meaning units the same initial codes. SS, SF and CL then discussed the initial coding in

**Table 1. Examples of meaning unit, code, sub-category, category.**

| Meaning unit | Code | Sub-category | Category |
| --- | --- | --- | --- |
| "I would rather trust people. . . and then I hope the police do their thing and arrest those criminals who cut our accounts and hurt us." | Trust in democratic society | Trust that personal data is kept safe | Trust and uncertainty about risks when using technology |

relation to meaning units, and made some adjustments to improve the coding. As part of the same discussion, we concluded that all data shared characteristics and was possible to consider as one unit of analysis even if the two rounds of focus groups were based on different constellations of participants. SS and SF compared the initial codes, focusing on the manifest content, based on differences and similarities and abstracted them to form sub-categories (Table 1).

Based on similarities and differences, we made further abstractions to form categories. The categorisation was repeatedly discussed with CL, and to ensure a valid and trustworthy description, the fourth (SMS) and fifth (SI) authors also reviewed and gave their input to the categorisation before the analysis was finalised.

Fig 1 displays the final categories and sub-categories. The first category mainly mirrors attitudes and experiences from participants in the two older generations, and its first sub-category (Technology development is a natural process) displays perspectives only from these two oldest generations (highlighted with lighter colour in Fig 1). The three remaining categories present attitudes and experiences expressed across all three generations.

## Findings

The four categories (Fig 1) illustrate generational—but also largely individual rather than generational—similarities and differences integrated in sets of sub-categories on technology and technology development. Table 2 provides an overview of the generational similarities and differences. Generally, the two older generations, more often than the 30–39 year olds, tended to express concerning views on the topic. The findings also reflect that participants across generations preferred to discuss a broad range of digital, rather than traditional technologies.

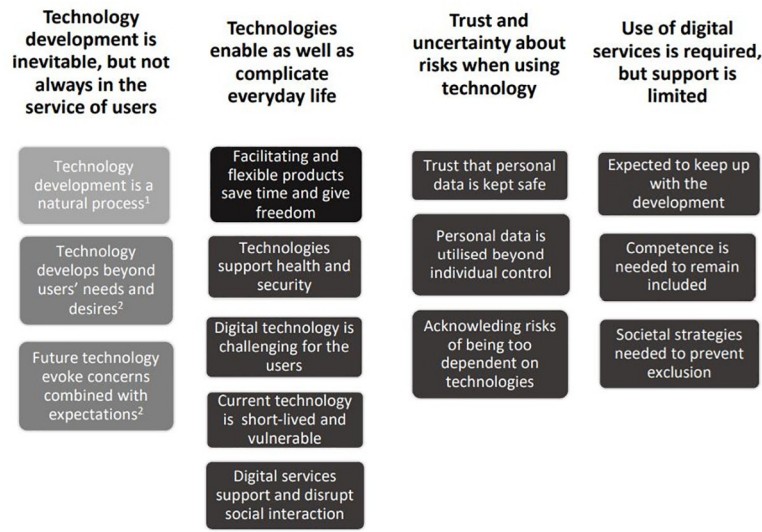

**Fig 1. Overview of categories and sub-categories, including a brief summary of generational differences and similarities.** [1] Sub-category representing only 50–59 and 70–79 year olds. [2] Sub-category representing mainly 50–59 and 70–79 year olds.

**Table 2. Overview of generational similarities and differences in the categories.**

| | Technology development is inevitable, but not always in the service of users | Technologies enable as well as complicate everyday life | Trust and uncertainty about risks when using technology | Use of digital services is required, but support is limited |
|---|---|---|---|---|
| Generational similarities: | The 50–59 and 70–79 year olds dominate discussions focusing largely on ethical perspectives in relation to technology development and future technology. | Generational and individual perspectives displayed. | All generations show some awareness of risks and uncertainty | General agreement across the three generations on necessity and challenges to keep up with development. |
| Generational differences: | The subcategories only and mainly represents 50–59 and 70–79 year olds (see Fig 1) | Interest for traditional rather than digital contact and challenges more pronounced with increasing age.<br><br>Strategies to deal with complications less pronounced with increasing age. | Degree of awareness and interpretation of risks varies; the 30–39 year olds seem more trustful and the 70–79 year olds more afraid of risks than those 50–59 years old. | Compared to those 30–39 the 50–59 and 70–79 year olds, expressed more concerns about risks of digital exclusion and struggled to keep up with digitalization. |

## Technology development is inevitable, but not always in the service of users

The first category (see Fig 1) focuses on technology development in the past, present and future, where digital innovations more or less over-shadow past innovations and technological shifts (e.g., TV, CD-recorder) experienced during the lifetime of the participants.

**Technology development is a natural process.** The two older generations discussed technology developments as natural and inevitable, as well as adjustments to previous technology rather than real novelties. They generally agreed that time helped to get used to technology: "We take quite a lot of technology for granted now, because we've had it for so long. And maybe a lot of things we talk about now, our grandchildren will think that we have always had." (W1, 70–79) (Code within parentheses representing sex, participant number and generation.)

Everyday technologies that have existed for a while (washing machines, stoves, refrigerators, etc.) were not even referred to as technology, or at least they were fully taken for granted because their configuration or use has not changed much. "If you consider a fridge from the 1930s, you turn on the power, open the door, and take out your goods, close the door and you would do the same today, nothing odd about that." (M4, 70–79). The 70–79 year olds acknowledged that technology developments had simplified activities in everyday life, reduced physically heavy household work and saved time across their life span–but concluded that development with a similar purpose now is poor.

**Technology develops beyond users' needs and desires.** This sub-category largely represents experiences displayed by the 50–59 year olds, for example, that products are developed based on technological possibilities rather than human needs: "that's smart, but makes it [the product] unmanageable. The technology itself is fantastic but to make it accessible or simple is unheard of." (W1, 50–59). Compared to when they were younger, they found themselves less trend-sensitive and surer of desired features in products such as washing machines: "new programs [doing same thing as others] aren't necessary. Those [programs] used all times work just great." (W5, 50–59). When acquiring new products, resemblance to previously owned products was considered important as well as costs, functionality and sustainability:

Yes, and I. . . have reached the age where you feel I want to take responsibility. I don't want a lot of stuff just standing there. Partly it's because it costs and I don't use it. I don't think

it's environmentally friendly. Why should you buy [something new] with just an extra small function on. (W3, 50–59)

The 50–59 year olds largely discussed intrinsic purposes of developing and introducing technology:

Does the technology have a purpose? Do we need to update all systems all the time just because we can? So we have to have an ethical discussion about what we are doing, what kind of society do we want? Just because we can do certain things does not mean that we should. This excludes people. (W1, 50–59)

In contrast, participants in the two younger generations argued that seemingly redundant innovations could develop into useful and desired technologies. They also acknowledged developments towards more user-friendly technology, as well as changes of own preferences over time;

I'm a technology freak and have been since I was a child. . .what impressed me before was the nicer and more remarkable functions. Now I become more and more impressed. . .You buy an expensive thing. . .press one bottom and it acts the way it should. . .There's not Star Trek panels so much as before, some stuff has become easier to use anyway. (M1, 50–59)

However, they did problematize that few persons in real need actually get access to such innovations: "This feeding robot exists and has been around for a long time. . . It's a really smart thing, but how many can use it?" (W1, 50–59).

Opinions differed among the participants about who (users or experts) should be involved in the development of new products. The 70–79 year olds identified themselves as a minority group and questioned why majority groups should have so much more impact on technology development. Nevertheless, participants across generations argued that user experience should be part of product and service development to generate more effective and sustainable solutions. Generational perspectives and age sometimes prevent such involvement: "The young people don't understand how, even 50 year olds think." (W2, 70–79), further exemplified by the following quotation:

It is difficult when you have reached this age to be involved in deciding. Maybe [you can] find channels [such as municipal user boards] and hope a little that you can influence. . .But I am not in a position where I have decision-making functions." (W1, 70–79).

**Future technologies evoke concerns combined with expectations.**   When reflecting upon future technologies, the 30–39 year olds found themselves overall optimistic. The 50–59 year olds were positive but also identified risks, and the 70–79 years olds found their generation to be more sceptical than the others, however, with individual differences:

I was about to withdraw from this [participation in the second focus group], but in my first group [70–79 year olds] I personally thought they were very negative towards the technology, and I felt a little worried—oh! Am I representative? No, I'm not. So I didn't feel like continuing because. . . I want to continue in the positive direction that I have. (W1, 70–79)

Generally, participants called for and expected simpler, more practical, smarter and more interactive technologies in the future. Products should demand less from users by making usage intuitive and guide them without requiring access to complex or extensive manuals.

Concerns were raised that the capacity of current housing (e.g., too few power sockets) might actually limit the use of technology and thus future possibilities.

Focusing on health care, development of new technologies such as those based on artificial intelligence (AI) was considered promising as machine learning could do a better job than humans, especially in diagnostics. The 50–59 year olds debated whether AI was adequately equipped to make ethical decisions (e.g., if self-driving cars would choose to hit objects rather humans in case of accidents). Participants across generations discussed the vulnerability of robots due to program flaws.

Around robots in care situations, opinions differed a lot and program failures and de-humanisation were risks mentioned. At the same time, the two older generations raised the risk of de-humanisation:

> I was at a mammography and it was organised similar to a car inspection. You had to log into a device, nobody welcomed you in, and you were a thing, a device. . . I chose not to understand [the instructions], to check what would happen and pressed the help button. Nothing happened, others came and logged in and were cared for and I sat there until someone finally said; who are you? (W5, 70–79).

While the younger generations seemed more positive to AI, the following conversation illustrates more concerns in the oldest generation:

- I want a living person that I can talk to. (W5, 70–79)

- Then we need to get people to work in those areas, but people are not so interested in working [in social and health care]. . .And I don't want to lie or sit somewhere and have no one at all to help me. (W1, 70–79)

- A robot coming in to feed me? (W5, 70–79)

- Yes, but it is better with the robot than lying there and neither getting food nor help to get out of my bed. (W1, 70–79)

- No [not for me] (W5, 70–79)

- I heard the opposite the other day actually. . .It was a woman who said "I would die if there was a man in my bedroom, then I would actually rather have a robot looking!" (W6, 70–79)

In discussions about other work life settings such as monotonous, industrial tasks, the use of robots did not raise any concerns at all.

### Technologies enable as well as complicate everyday life

The second category (see Fig 1) elucidates that technologies that are self-explanatory, user-friendly, easy to use, equipped with fewer buttons or functions, require less effort and have fewer choices to use were desired across generations. Described in more detail in four sub-categories, participants choose to use everyday technologies and ICT (e.g., cars, household appliances, computers, smart phones TV, radio, game consoles), but were sometimes challenged when using them.

**Facilitating and flexible products save time and give freedom.** All generations saw the smart phone as the most important product given its multi-functionality, being a combined diary, photo album, dictionary, map, wallet, etc.: "a swiss army knife. . ., including it all. . . you can do everything, even visit the doctor." (W5, 50–59). On top of these multiple functions, the

smart phone brought possibilities to control the environment, used by participants in the two younger generations. Furthermore, technology use contributes to increased freedom and flexibility in participants' private, social or working life: "I respond [through my smart phone] when I have time and I send [messages] when I have time. . . So that you allocate your time better I think with technology that has come." (W1, 70–79). Digital meetings save time by reducing the need to travel (at work or within senior organisations). The participants debated whether digital payments (e.g. the Swedish payment app called Swish) were safe and secure, resulting in individual rather than generational decisions to avoid or accept. When it comes to communication modes, the 70–79 year olds generally argued for traditional rather than digital contacts, but saw the advantages of digital text messages that automatically generate proof of discussions compared to phone calls in consumer complaint situations.

**Technologies support health and security.** The participants generally agreed that assistive technologies for communication (e.g., alternative communication devices, hearing aids), mobility (e.g., powered scooters, wheelchairs, wheeled walkers) or other use support independence and thereby health. They also appreciated medical devices (pace-makers, hip replacements, etc.) supporting health and survival. The younger generations utilised digital primary health care units, especially regarding their children. However, they avoided sleep monitoring arguing that sleep patterns are hard to change, but otherwise found health and activity monitoring applications (apps) and devices helpful to support a healthy lifestyle and security:

> If I fall [due to vertigo], the watch will feel it. . .and asks if I want it to call my wife or to 112. . .If I don't respond . . . it calls my wife and 112 and sends my position and it's just a consumer product. (M1, 50–59)

Similarly, access to landline or smart phones for contact in case of emergency at/out of home gave the oldest generation a sense of security similar to a personal security alarm. The younger generations appreciated similar benefits related to older family members or smaller children.

**Digital technology is challenging for the users.** The number of digital products available, the necessity to integrate them in the home setting, as well as their complexity in design and abstract functions challenged all participants:

> Now there is a lot of technology that is used, and maybe only five people in the world know exactly how it works. . .Technology doesn't get any less advanced, it may not be possible to get rid of it or so, but you could think more about how to simplify things or make things a little more transparent. (M1, 30–39)

Participants did not always appreciate when products made decisions for them, and especially the older generations asked for simpler products (e.g. washing machines) that allowed manual choices: "In the past, you could set the degrees on one button and then select the program on another. Now these functions are linked, you can't wash 90 degrees using a short time program." (M4, 70–79).

For all generations, digitalisation sometimes complicates service provision and causes frustration when apps fail, need to be updated or downloaded; codes need to be acquired, are forgotten or do not work. Sometimes multiple lengthy phone calls are needed to sort out the problems. To avoid such complications or spamming, participants across generations sometimes restricted their use of digital technology–considering it a relief to do without the smart phone.

**Current technology is short-lived and vulnerable.** Generally, participants believed that current technologies have lower quality and shorter life expectancy than previous products. Pushes to buy new products were disliked: "a phone with a subscription usually has 24 months of binding time, and then they try to get you to sign a similar subscription with a new phone" (M1, 30–39). All generations generally disliked such market driven consumption of short-lived, often unrepairable, technology that consumes money and time, and the two younger generations also found this unsustainable. The 30–39 year olds updated single digital products in order to better support desired outcomes or used strategies to increase the lifetime of products:

> I have had my iPhone for almost 4 years and . . .I try to have less on the phone, put it [information] on the computer, clear notes, and clear text messages from people you do not talk to, clear contacts, and try to update and use it a little less. It's really difficult, but it [the smartphone] recovers. (W2, 30–39)

However, this generation argued that consumer expectations were sometimes too high: "The mobile phone. . .is basically a whole laptop. . .in my pocket and I sweat and it rains, and it's hot and it is cold, so if it dies after 2 years isn't strange." (M2, 30–39).

Across generations, participants discussed how digitalisation made everyday technologies more vulnerable: "Everything has some functionality that relies on software . . . also refrigerators. . .and every time they update the software it requires more of the hardware as well." (M1, 30–39). The price-quality relation was yet another issue debated across generations: "If you only knew it lasted longer. . .I would be willing to pay a little more, am I being fooled? Or is it actually better as well?" (W1, 30–39). The 30–39 year olds concluded that regular consumers are not qualified to assess either the quality of today's advanced products, or whether it is worthwhile to invest in new, advanced technologies.

**Digital services support and disrupt social interaction.** The two older generations considered digital socializing more superficial than traditional. They also largely agreed that the integration of digital products sometimes disrupts in person social interactions: "even if I am so positive about phones, I do not want to sit and eat a meal and talk [with others who] do not focus on us sitting here." (W1, 70–79). Similarly, the two older generations found social media notifications distracting to the user and disruptive of other activities or interactions. Strategies to adapt, restrict or avoid social media were dependent on individual considerations of risks and privacy preferences beyond generation: "I use Facebook and I'm positive about it." (W1, 70–79) and "I found it interesting in my group [30–39 year olds], . . . that so many weren't on social media and . . . didn't want to be part of it." (W3, 30–39). Generally, participants accepted digital meetings (at work, within organisations) as a means to reduce travelling, but the two older generations preferred personal physical meetings because they facilitated social interaction.

## Trust and uncertainty about risks when using technology

In the third category (see Fig 1), participants showed awareness about personal integrity risks pertaining to digital technology use. On the one hand, they were generally trustful that their personal information was kept safe; on the other hand, they were uncertain about what such risks entailed and how to understand them.

**Trust that personal data is kept safe.** The 30–39 year olds were generally trustful: "We have a good situation in Sweden. I myself have quite a lot of confidence in the government and the state, and I don't think they will use my information badly." (M3, 30–39). However, some

participants experienced scepticism about getting offers based on previous internet searches: "Just because they [commercial companies] say this is what matters to and defines me, it doesn't have to mean it is right. I often think the personalised advertisements you get are not right at all." (M3, 30–39). The two older generations debated trust and user naivety, exemplified by the following dialogue:

- I would rather trust people. . . and then I hope the police do their thing and arrest those criminals who cut our accounts and hurt us. (W4, 70–79)

- It's probably not good to have the location service [on the mobile phone] running all the time and people always know where you are. . . (W2, 50–59)

- I think the benefits outweigh the risks. . . Somehow; they [surveillance cameras] don't bother me when I walk in the community because I have nothing to hide. And those who may have something [to hide] . . .may be good to catch on camera. (W1, 70–79)

The 50–59 year olds displayed trust that information logged from smart phones/watches was stored anonymously, and merely used for common good, such as long-term prediction of health or for traffic guidance (i.e., for rush hours and accidents). However, participants across generations were sceptical towards what would happen if insurance companies or other service providers accessed such information. Upon own approval or initiative, the younger generations were open to share health information from their private digital equipment with health care providers. In case of emergency, participants across generations were positive about sharing GPS data with emergency services without explicit consent. The 50–59 year olds concluded that using welfare technology such as night cameras entailed lower risks for care recipients than to let unknown staff into their homes.

**Personal data is utilised beyond individual control.** While the 70–79 year olds did not fully trust digital patient records because medical staff not involved in the patient relationship could easily access information, the 50–59 year olds argued that unauthorized access of such data was traceable. The 30–39 year olds suggested that it should be easier to exchange digital health records across organisational borders.

Participants agreed that security is less of a problem under "normal circumstances". However, they saw risk of someone obtaining information for unwanted political or criminal activity. The 30–39 year olds argued for individual responsibility to protect personal information.

I sometimes make a conscious decision not to share [information on social media]—I don't have Facebook so my kids aren't on Facebook. . .It doesn't matter how careful I am in all other media because I have it [the smart phone] in my pocket. (M2, 30–39)

While the two older generations suggested authorities (e.g., the Swedish Tax Agency) should automatically protect their citizens. The 70–79 year olds experienced this to be out of their hands;

I'm afraid of the abuse that can happen because the tech nerds . . . control the development. . .We [my age group] don't have enough knowledge, in general, to perhaps analyse risks that way—while your generation [talking to a person 30–39 years old]. . . When we [my children and I] discuss such issues, they come with completely different aspects. I then think—oh, maybe I should be afraid of it? I don't know. (W1, 70–79)

The oldest generation believed that younger people ignore risks while "many older people are afraid of new stuff. They don't want to know, it's nice not to know" (M3, 70–79). Some participants applied systematic strategies (e.g. deleting web cookies regularly) to avoid tracking, and some were hesitant towards buying things online. They argued for increased restrictions on how companies are using the vast amount of individual data they are collecting because current legislation, such as the General Data Protection Regulation (GDPR) within EU (that directs who is allowed to sell personal information), is not sufficiently respected.

**Acknowledging risks of being too dependent on technology.** All generations discussed dependence on technology as a potential personal risk, and the 30–39 year olds identified the same risk for their children. Being dependent on reminders was one such risk, illustrated by this dialogue:

- Be careful not to get confused when you get older. . .As you get older, it's a sport to remember what to do during the week, without looking at the calendar. (M4, 70–79)

- . . .Well, I'm probably training the brain more now than before because—it depends on how you are as a person, I'm pretty curious and as soon as I come across something I wonder about, I pick up the phone and look it up immediately, and I learn something about it. . .So I guess I keep my brain going. (M1, 50–59)

Beyond reminders, participants across generations found dependence on digital technologies to buy and keep tickets, guide routes, etc. problematic and demanding due to risk of failures: "If the computer breaks down, you are totally lost. You can't pay the bills, you can't do anything." (W2, 30–39).

The two older generations were especially concerned about the lack of preparedness for failures due to natural or enemy-forced system breakdowns, "I feel more and more vulnerable for every week . . . when I hear things [incidents] on the radio." (W5, 70–79), especially concerning welfare technology or vital societal functions or specifically:

What happens when it doesn't work as we expect. . . Now all of Sweden goes down if an electric cord goes off. This shouldn't happen. But because they've been so eager—to get this out, fast, out to everyone. . . these errors come. Yes, we must find out. . . the errors in advance. (W4, 50–59)

## Use of digital services is required, but support is limited

In this last category, the participants across generations acknowledged challenges of varying sort and degree to keep up with the rapidly emerging digital technologies. They also discussed the general lack of support important to avoid digital exclusion in especially vulnerable populations.

**Expected to keep up with development.** Similar to other norms that maintain a society, participants accepted that individuals generally have no other choice than to use digital technology. Nevertheless, in the context of digitalisation, this lack of choice brought individual problems for vulnerable groups (e.g. older adults with limited digital competence), because tailored support is lacking:

You can place a responsibility on the individual if the individual is capable of taking that responsibility. . .Now if you can't, it's your [the individuals] problem. . .We have a norm

and we have a policy. . .It should look the same to everyone because it is a fair thinking in the distribution of resources and not in the outcome." (W1, 50–59)

Beyond ageing, participant's perceived that digital solutions, more often than traditional systems, transfer the responsibility for solving problems to the service user:

When arriving at the first parking lot, the ticket machine was completely off. When I came to the other, it was out of order. I came to the third that only allowed payment by an App. Then I called the parking company and thought they would say I can park for free. . .They said I have to download that app, otherwise you have to go to another place. (M2, 70–79)

They also considered it challenging to have no other choice than to deal with the deficiencies of new, prematurely released technologies and to keep up with the development although users were not sufficiently prepared, "You can't sit down and consider, do I need this. . .can I manage without it? You have to choose otherwise you can't pay any bills. . . You can't stop and think because everything goes so fast." (W5, 50–59).

**Competence is needed to remain included.**   Generations agreed that digital competence and knowledge are key components to become or remain included. The 30–39 year olds did not seem to question their competence, while the 50–59 year olds considered themselves competent enough and the 70–79 year olds considered themselves to be less skilled. However, they acknowledged experiences in areas that younger generations lack: "There's some fun stuff on the internet, on YouTube. Where you see a boy trying to use an old-fashioned phone. He has never seen anything like this before, and can't find out how to use it." (M3, 70–79). When acquiring new technologies, trial and error is often the only option to get it working for the 70–79 year olds: "But there is also a certain fear, oh, what if I do wrong, what if I ruin things now. . .It's not easy." (W1, 70–79).

Participants across generations considered the ability to assess risks and accuracy of digital information as vital competencies, beyond being physically or cognitively able to handle the devices. If such digital competencies are lacking, support from service providers is vital to avoid exclusion.

**Societal strategies needed to prevent exclusion.**   Participants in the two older generations argued that society should not only save lives, but also support a good life in case of disability, "If they [authorities] give us assistive devices, they also have to adapt the society for these assistive devices." (W5, 70–79). They considered that authorities or service providers offer no support relating to their digital services. Instead, municipal libraries, local bank companies, senior organizations, and above all younger family members, provide support according to the two older generations. Because support is scarce, particular older adults living alone or without family members nearby have difficulties finding help. All generations called for better and more flexible support services, access to human support and overall resources, including manuals adapted to the needs of different target groups.

Participants in the middle generation suggested that social services should provide support and essential ICTs on equal terms such as for welfare technology or assistive devices, at least to people with limited financial resources. The two older generations considered parallel systems important to avoid exclusion:

I think it's great to use all the new technology, for my part it could have developed faster, but. . .traditional methods. . .must remain as long as the target group and the users remain. Or a reasonable transition period or plan or something is in place. (M1, 50–59)

## Discussion

The present study shows that people across generations may be largely unanimous in viewing technology as an intrinsic and vital aspect of their daily lives. Even if the aim of the study was to focus on different types of technologies, participants clearly highlighted digital technologies as the "fundamental" expression of technology today, supported by the fact that the wide range of technologies considered in the present study increasingly involve digital components (eg. smart appliances or self-driving cars). Nevertheless, previous research rarely displayed what technology and particularly emerging digital technology use (including ICTs) truly entails within and across generations. Importantly, technologies and technology development have implications, for better and for worse, resulting in attitudes along continuums reflected by our findings (Fig 1). Although these better-worse continuums were evident across the generations included in this study, the two older generations tended to pronounce concerning views (closer to the worse end) to a larger extent than the 30–39 year olds.

Across the continuums of implications for better and for worse, individual attitudes shaped by factors such as personality, interests and feelings (e.g. degree of trust, craving change vs. stability; being interested, neutral or outright disinterested in new gadgets, etc.) rather than chronological age or generation seem to account for the participants' attitudes regarding technology. In fact, individual attitudes that shape use of technology may prevail regardless of age [27]. Again, supporting the notion that generations need to be redefined over time [10]. Nevertheless, age coding attitudes prevail in social encounters [28] as avoidance to use certain technologies (e.g. digital payments such as Swish) in a 35-year-old is likely to be interpreted as a personal choice while a 75-year-old making the same choice will be considered to not understand the technology or be tech-unfriendly and less open to change. The influence of individual attitudes rather than generational attitudes in the area of technologies and technology development is a finding rarely described in research, but warrants investigation through research designs allowing for generalization (e.g. the forthcoming survey, which is part of the GenerationTech project). Meanwhile, and given the implications of age coding discourse on, for example, social inclusion [28], this is a message necessary to consider in policy-making and service design.

Although technology brings flexibility, people across generations sometimes feel forced to use digital technology, as parallel systems are lacking or failing and digital technology is an increasing part of the infrastructure of society [29]. Digital service provision is sometimes pursued ad absurdum, which implies it is mandatory to carry a mobile phone (irrespective of whether you want or are able to use such devices) as in the example told by one of our participants forced to pay a parking ticket by mobile app when the parking meter broke down. On the one hand, technology saves time and brings accessibility, flexibility, autonomy and a sense of security to people. On the other hand, insecurity (due to e.g. digital hacking, data mining, failure or not wholly understanding said technology) is also part of the picture. Similar to recent research [30] participants in the two oldest generations were sceptical of private technology companies and their focus on the economic bottom line, as well as their lack of regard for ethical aspects such as exclusion, integrity or privacy.

Our findings indicate that people aged 30–39 years accept more individual responsibility, people aged 70–79 years generally want the society (authorities, service providers, etc.) to support the citizens, while people aged 50–59 take a position in between—accepting such responsibility yet arguing for official support for groups at risk of lagging behind. The aftermath of technology change and what it entails for the individual is debated, for example, considering eHealth, which is perceived to transfer responsibility from health providers to the individual in a simultaneously empowering and disempowering fashion [31]. For better and worse, the

responsibility of the individual over societal/government domains has increased since the middle of the 20[th] century world-wide in line with the emergence of neo-liberalism [32]. The attitudes revealed by the present study are interesting and warrant further investigation.

In line with the results of a recent study from Sweden [11, 33], generally the younger generations seem to fall more easily into a category of self-reliant "digital natives" [11], while the oldest generations, despite also utilizing numerous types of technology, express less confidence in themselves. The 30–39 year old participants in our study revealed more homogenous attitudes of technology compared to the two older generations. This may be due to the particular constellation of personalities, homogenous backgrounds, unwillingness to voice more dissenting opinions, or difficulties to criticize technology they feel so "close" to (having experienced fewer technological transitions than the older generations) [34].

However, across generations there was also broad agreement, for example, defining the mobile phone as the most important gadget, including "Swiss army knife" functionality [35]. Mobile phones are described as an extension of ourselves [36] carried around on our bodies in a highly mobile sense [35], and are nowadays difficult to do or function without [37]. Nevertheless, the generational agreement found in our study on these issues has rarely been reported or highlighted in previous research.

Whether and how current technology development satisfies vital human needs was introduced by a female participant in the middle generation as a broad discussion of technology-push and demand-pull [38] including technological determinism, that is, "the technology of a society determines the development of its social structure and cultural values" [39, p 727]. It must be remembered that focus groups differ from individual interviews, as some opinions (yet relevant) do not stem from the entire group [40]. As none of the other participants argued against the issues raised by this single participant and the two older generations (rather than the 30–39 year olds) largely supported her viewpoint, our interpretation is that similar opinions based on ethical considerations may grow as people age. Such ethical dialogues are largely missing in mainstream technology research as well as in technology development processes, but they certainly called for. For example, it would be relevant to consider the debate among the 70–79 year olds on AI technology as a substitute or supplement to human contact in the development and provision of social care for older people. Participants who considered it a supplement were also more open to receive help from robots rather than stressed staff. In line with the results of recent research [41], privacy and personal autonomy influence the use and acceptance of technology. Over time, new and more controversial technologies may also bend attitudes towards previously questioned novelties.

Nevertheless, some actions are vital to drive the development towards the "better-end" of the better-worse continuum. Researchers and policy-makers should pay attention to the desire for continued parallel systems of technology, preserving well-established technology alongside advancements in order to prevent digital exclusion [33]. Policy-makers should also consider integrity and privacy of citizens regarding technology to enable inclusion in the increasingly digital society, as also suggested by [42]. Support for understanding and troubleshooting new and existing technology should be designed for different target groups and made fully available regardless of chronological age, as such support would increase inclusion and self-confidence among older users [41]. Supported by our findings, younger generations would also benefit from information to understand and deal with hacking or "scamming" risks on public Wi-Fi, online services, social media, etc. Moreover, our participants called for societal plans that would protect citizens in case of more widespread and lasting power outages or other "digital disasters" disrupting everyday activities dependent on digital technologies. Moreover, it is imperative for decision makers to adjust, renovate or improve infrastructure along with

technological developments in order to increase technology usability and reduce exclusion across generations.

For future technology development, the three generations shared some basic desires, for example, longer product life spans (accepting some higher initial cost) and transparency of features and quality in more advanced products, not wanting to feel "scammed" by a purchase. Accordingly, technology companies should not only gather information to personalise their marketing efforts, but also identify and consider users' needs and desires for upcoming technology. The participants appreciated the ability and simplicity to use new technology right away, without requiring active hands-on actions from the user. This negates the idea that only older users need simplified technology, and supports the notion that when usability improves for one, it improves for all [43]. Taken together, such attitudes are highly likely to decrease the more negative part of and implications along the better-worse continuum for individuals across generations.

## Strengths and weaknesses

This study is a part of the paradigm shift from research approaches viewing older adults as a homogeneous group with identical needs, towards a more in-depth, personalized approach with increased user influence. Historically, users of technology—and older users in particular —have lacked interpretive precedence and in many contexts are still invisible [22, 23]. The strength of the current study is that it sheds light not only on the current population of older adults, but also on current attitudes in future older populations. In addition, the study design including first age homogenous and thereafter age heterogeneous focus groups made it possible to highlight not only differences between but also within generations, as well as similarities in attitudes across generations rarely identified in previous research.

The fact that the majority of the participants had an upper level or university education likely influenced our findings. Moreover, men were somewhat underrepresented in the middle generation (50–59 year olds), and the perspective of 30–39 year old men was missing in the second round of focus groups due to a same day cancellation. Additionally, an overrepresentation of "tech savvy" persons, particularly among the men, was evident according to the researchers moderating and observing the sessions. On the other hand, the topic of the study was technology and a certain level of interest in the subject was key to enable rich discussions.

Given the design, the findings can to limited extent be transformed beyond the study sample, but findings are transferable to similar settings as participants of different ages, demographics and backgrounds were recruited, and rich descriptions were generated to enhance the transferability of the findings [44]. As we have used the findings from the present study for the development of questions for the survey in the GenerationTech project, the results from that ongoing study will further validate the present findings.

The focus group methodology was relevant for capturing experiences and attitudes in diverse groups and to achieve engaging discussions towards the study aim, and the credibility of the findings is supported by prolonged engagement and member checking included in the second round of data collection. Continuous discussions between the researchers mainly involved in the analyses (SF, SS and CL) and the use of NVivo for data analysis supported dependability, and the two remaining authors (SMS and SI) provided peer debriefing to further validate and confirm the findings [44].

## Conclusion

Even though younger people reveal more homogenous attitudes toward technology compared to older people, experiences of and attitudes towards technologies and technology

development are not limited to generation; perspectives sometimes unite individuals across rather than within generations. Thus future technologies and technology development, as well as services and policies aiming to support the use of said technologies should consider individual user perspectives including needs, desires, beliefs or goals neglected in the existing technology models, and involve users beyond generations defined by chronological age. Such strategies are likely to be successful in supporting development of technologies usable for all.

## Acknowledgments

We would like to thank the study participants for their contributions. The present study was conducted at the Centre for Ageing and Supportive Environments (CASE) at Lund University.

## Author Contributions

**Conceptualization:** Charlotte Löfqvist, Steven M. Schmidt, Susanne Iwarsson.

**Data curation:** Sofi Fristedt, Samantha Svärdh, Charlotte Löfqvist.

**Formal analysis:** Sofi Fristedt, Samantha Svärdh.

**Funding acquisition:** Charlotte Löfqvist, Steven M. Schmidt, Susanne Iwarsson.

**Investigation:** Sofi Fristedt, Samantha Svärdh, Charlotte Löfqvist.

**Methodology:** Sofi Fristedt, Charlotte Löfqvist.

**Project administration:** Sofi Fristedt.

**Supervision:** Charlotte Löfqvist, Steven M. Schmidt, Susanne Iwarsson.

**Writing – original draft:** Sofi Fristedt.

**Writing – review & editing:** Samantha Svärdh, Charlotte Löfqvist, Steven M. Schmidt, Susanne Iwarsson.

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
