## [Decision Letter · Decision Letter 0]

14 Jan 2021

PONE-D-20-32347

Am I representative (of my age)? No I'm not - Attitudes to technologies and technology development unite individuals across rather than within generations

PLOS ONE

Dear Dr. Fristedt,

Thank you for submitting your manuscript to PLOS ONE. After careful consideration, we feel that it has merit but does not fully meet PLOS ONE’s publication criteria as it currently stands. Therefore, we invite you to submit a revised version of the manuscript that addresses the points raised during the review process.

The manuscript deals with an important research topic regarding possible exclusion due to low adoption of technology in a more and more digitalized society. It develops several important points to consider as potential barriers and challenges in the use of technology.

I agree with the points raised by the reviewers, particularly about embedding the current study more into the existing literature on technology use. Since a focus of the study is on generations and age effects, I would suggest to align the study as well with scholarship on technology use at older ages. I encourage you to define more clearly your subject of study - technology is currently defined too vague - and relate your findings more to existing scholarship particularly coming from psychology, on individual characteristics related to attitudes, beliefs, and motivations. Please find additional comments below.

We look forward to receiving your revised manuscript.

Kind regards,

Anja K Leist, Professor Dr.

Academic Editor

PLOS ONE

Journal Requirements:

2.Please provide additional details regarding participant consent. In the ethics statement in the Methods and online submission information, please ensure that you have specified (1) whether consent was informed and (2) what type you obtained (for instance, written or verbal, and if verbal, how it was documented and witnessed). If your study included minors, state whether you obtained consent from parents or guardians. If the need for consent was waived by the ethics committee, please include this information.

3.We note that you have indicated that data from this study are available upon request. PLOS only allows data to be available upon request if there are legal or ethical restrictions on sharing data publicly. For information on unacceptable data access restrictions, please see http://journals.plos.org/plosone/s/data-availability#loc-unacceptable-data-access-restrictions.

Additional Editor Comments:

In general, it would be helpful to embed the rationale and conclusions from the study more into existing models of technology use, see for example a recent review here:

Abri, D., & Boll, T. (2020). Aging, technology, and psychology: Models of assistive device use viewed from an action-theoretical perspective on lifespan development. European Psychologist, 25(3), 211.

Since the topic of the study is quite large, it may be necessary to go to subdomains of technology, such as ICT use, computer use, or use of assistive devices.

A description of participants in terms of education would be helpful in the methods section, in line with what the reviewers mention.

Finally, the "individual-minded" concept that is used particularly in the discussion is rather vague and hard to understand. It would probably make more sense to replace this concept with more established factors from psychological research on, e.g. openness to and positive attitudes towards technology use, motivational factors in technology use, from, again, existing models of use of technology, as the reviewers suggest. The conclusion would then be that these individual predispositions rather than age determine attitudes towards and acceptance of technology. That would also link the present study better to the well established finding that age is often not a relevant predictor for technology use.

- the following terms or sentences were hard to understand or suspected wrong, please consider replacing/rephrasing:

p. 3 surf tablets

p. 3 artificial body organs

p. 9 CD-recorder

p. 14 participants choose to be without products

p. 14 proc

p. 14 communicators

p. 23 more or less no support

p. 24 better-worse continuums rang

p. 24 age coding attitudes

p. 25 data mining - replace by collection of data?

Reviewers' comments:

Reviewer's Responses to Questions

**Comments to the Author**

1. Is the manuscript technically sound, and do the data support the conclusions?

Reviewer #1: Yes

Reviewer #2: Yes

2. Has the statistical analysis been performed appropriately and rigorously? 

Reviewer #1: N/A

Reviewer #2: N/A

3. Have the authors made all data underlying the findings in their manuscript fully available?

Reviewer #1: No

Reviewer #2: No

4. Is the manuscript presented in an intelligible fashion and written in standard English?

Reviewer #1: Yes

Reviewer #2: Yes

5. Review Comments to the Author

Reviewer #1: The study focuses on the experience of participants from three generations about use of a broad range of technologies and their attitudes toward technologies. The authors formed four categories regarding the attitudes of participants toward technologies. Via qualitative analysis, the authors concluded the essential role of individual user perspectives from different generations for the acceptance or use or development of technologies.

I have several comments regarding different parts of the manuscript that follow below.

Note.

Following page numbers refer to the numbers at the bottom of pages in the manuscript.

Line numbers refer to the line numbers of the document.

Title

1)The title is limited to a verbatim citation of one participant. I recommend that the authors modify it to a title that readers can find what follows generally in the study. My suggestion is something like “Attitudes towards use of technologies in three generations: A qualitative study”.

2)I recommend that the authors consider some more keywords that cover the major contents of their paper. None of the three keywords that have been mentioned contain the main terms in the title too. I recommend to add some more keywords that refer to the title and main points in the paper.

Introduction

1)Please bring a reference for “While technologies are broadly defined as goods, services, knowledge, skills etc. (p2, line 23).

2)Please bring a reference for four categorizations regarding technologies (p3, line 5) (or are they categorized by the authors? How did they categorize them?

For instance, the authors categorized eHealth solution as a welfare technology, they can be categorized as assistive technologies or as information and communication technologies as well. Or the authors categorized TV as everyday technologies, it can be categorized as information and communication technologies too. Thus, as there is a different categorisation for various types of technologies, a scientific source about how technologies have been categorized is required.

Method

1)The authors should bring a reason in a clearer way about why they selected such method.

2)Inclusion and exclusion criteria for composing a sample or key characteristics of the sample should be more clarified. The authors should make explicit why they compose such sample (what are the criteria for collecting such sample). This information may help other researchers if they want to replicate a similar study.

Findings

1)As authors consider four categories for technologies, the findings about attitudes or experiences about use should also be clarified regarding different types of technologies and their category (most verbatim citations that authors brought in this paper refer to information and communication devices).

It is an important point because technologies in different categories have specific attributes, then the attitude towards each category may be different for instance the use experience of assistive technology such as wheelchair can be different in comparison with mobile phones. Thus, it may arise different concerns.

This distinction regarding different categories of technologies should also be illustrated in Figure 1 (It is not clear the term of technologies refer to which four categories of technologies that have been mentioned by the authors). Digital technology is just one that is clear in Figure 1.

2)Preparing the main findings in a form of table regarding attitudes' similarities and differences in three generations help readers to grasp the message easier.

Discussion

1)In the last line of page 23, it is mentioned that the wide range of technologies considered in the present study increasingly involve digital components (please bring some examples of these technologies that involve digital components and have been considered in this paper).

2)“The presence of individual-minded rather than generational attitudes in the area of technologies and technology development is a finding rarely described in research” (P24, line 17).

This point is an important point that can be more elaborated by the authors as there are some influential models about technologies that consider individual user perspectives. The authors may emphasis which individual perspectives they discovered in their paper (please bring some examples maybe beyond the individual user factors that have already been considered in the existing models).

Conclusion

1)The authors concluded that “future technologies, as well as services and policies aiming to support the use of them should consider individual user perspectives” (p29, line 15).

As discussed in the comment above, The point regarding individual user perspective is an important point that needs to be more elaborated by bringing some examples (which aspects of individual user perspectives have not been considered in the previous model studies or in the previous qualitative studies that should be considered in the future).

In the discussion part, it is mentioned (page 24) that personality and feelings are two important things to be considered. These two factors have already been considered in some existing models of technologies.

Some examples for individual user perspective (that the authors discovered by the current study) are required here as examples.

Reviewer #2: This is an interesting paper. Exploring the differences and similarities in attitudes towards digital technology across different age cohorts is a timely and important subject. The paper is well written and structured; the research question is clear and reasonable; the methods are appropriate; the findings are interesting. I acknowledge the authors for using a qualitative approach, which may potentially contribute to better understanding the thoughts and processes when dealing with digital technology. However, some issues came up in my reading, mainly having to do with the theoretical and conceptual rationale of the study:

1) After reading the introduction, it was not entirely clear to me what is missing in the literature and how this research could answer this gap. The authors state that few studies have used qualitative methods and cross-generational perspectives but I would appreciate a more specific theoretical and methodological rationale clarifying the overall importance and novelty of this particular study.

2) Following up to this point, I am missing a rationale for the relevance of the overarching research topic. The authors might consider shortly discussing the role of digital technology for enhancing active and healthy aging, for example, with respect to cognitive functioning, depression or social support (e.g., Cotton et al., 2014; Czaja et al., 2017; Kamin et al., 2020).

3) The authors should explain and justify their methodological approach. There are many types of qualitative methods and I would like to know why they have focused on qualitative content analysis to address their research question.

4) The authors correctly argue that technology attitudes are a complex research issue. There are many factors (e.g., cognitive, physiological, motivational, social) associated with technology use across the lifespan and therefore I was wondering whether and how this heterogeneity was reflected in the sample. If possible, it would be helpful to have more information about the sample (e.g., health status or functional limitations, previous experience with technology).

5) If possible, please provide the inter-rater reliability for the categorization. This would greatly support the methodological consistency of the paper.

6) After reading the discussion, it remained unclear how the findings contribute to the existing literature in the field. What does this study contribute to the key issue of interindividual differences in (late life) technology adoption/acceptance?

Literature

Cotten, S. R., Ford, G., Ford, S., & Hale, T. M. (2014). Internet use and depression among retired older adults in the United States: a longitudinal analysis. The journals of gerontology. Series B, Psychological sciences and social sciences, 69(5), 763–771, Article Journal Article. https://doi.org/10.1093/geronb/gbu018

Czaja, S. J. (2017). The Role of Technology in Supporting Social Engagement Among Older Adults. Public Policy & Aging Report, 27(4), 145-148. https://doi.org/10.1093/ppar/prx034

Kamin, S. T., & Lang, F. R. (2020). Internet Use and Cognitive Functioning in Late Adulthood: Longitudinal Findings from the Survey of Health, Ageing and Retirement in Europe (SHARE). J Gerontol B Psychol Sci Soc Sci, 75(3), 534-539, https://doi.org/10.1093/geronb/gby123

6. PLOS authors have the option to publish the peer review history of their article (what does this mean?). If published, this will include your full peer review and any attached files.

Reviewer #1: No

Reviewer #2: No

---

## [Author Response · Author response to Decision Letter 0]

21 Mar 2021

Response: Thank you for this comment. The style requirements have been checked and the manuscript is changed accordingly.

Response: Information on informed consent is now provided on page 6, line 2 as follows; Additional information about the study was then provided, and verbal consent to participate in the study was established. At the start of the first focus group session, all participants signed written informed consent. 

The study included no minors.

Response: Data availability The data used in this study contains sensitive information about the study participants and they did not provide consent for public data sharing. The current approval by the Regional Ethical Review Board in Lund, Sweden (No. 2012/558) does not include data sharing. A minimal data set could be shared by request from a qualified academic investigator for the sole purpose of replicating the present study, provided the data transfer is in agreement with EU legislation on the general data protection regulation and approval by the Swedish Ethical Review Authority. 

Contact information:

Department of Health Sciences, Lund University

Box 157, 221 00 Lund, Sweden

Contact address: DHSdataaccess@med.lu.se

Principal investigator: susanne.iwarsson@med.lu.se

Swedish Ethical Review Authority, Box 2110, 75 002 Uppsala, Sweden. 

Phone: +46 10 475 08 00.

Reviewers comments and author’s responses

Comment: In general, it would be helpful to embed the rationale and conclusions from the study more into existing models of technology use, see for example a recent review here:

Abri, D., & Boll, T. (2020). Aging, technology, and psychology: Models of assistive device use viewed from an action-theoretical perspective on lifespan development. European Psychologist, 25(3), 211.

Since the topic of the study is quite large, it may be necessary to go to subdomains of technology, such as ICT use, computer use, or use of assistive devices.

Response: Thank you for this comment. We have considered it and added the following text on page 4 line 8: In addition, most of these originate from disciplines such as information system engineering, business or management science. They neglect or only implicitly consider desires, beliefs and goals of potential users (21). In contrast but in line with the underpinnings of gerontechnology, interdisciplinary efforts integrating multiple disciplinary and user perspectives are certainly called for to address this neglect (14). In fact, few studies have adopted qualitative, user-centred research designed to create a deeper understanding of attitudes toward technologies, especially in the light of differences and similarities between and across age cohorts or generations.

Comment: A description of participants in terms of education would be helpful in the methods section, in line with what the reviewers mention.

Response: Descriptive data on education was presented already in the original submission, see page 6, line 8. We have added some more details the same paragraph: Ten lived alone and the rest were cohabiting. A majority of the participants reported being in good or very good health. Together they represented a diversity of professions.

Finally, the "individual-minded" concept that is used particularly in the discussion is rather vague and hard to understand. It would probably make more sense to replace this concept with more established factors from psychological research on, e.g. openness to and positive attitudes towards technology use, motivational factors in technology use, from, again, existing models of use of technology, as the reviewers suggest. The conclusion would then be that these individual predispositions rather than age determine attitudes towards and acceptance of technology. That would also link the present study better to the well established finding that age is often not a relevant predictor for technology use.

Response: Thank you for this relevant comment. We have taken away the word “minded”, and also added that attitudes are shaped by different factors. The text on page 26, line 19 now reads: Across the continuums of implications for better and for worse, individual attitudes shaped by factors such as personality, interests and feelings (e.g. degree of trust, craving change vs. stability; being interested, neutral or outright disinterested in new gadgets, etc.) rather than chronological age or generation seem to account for the participants’ attitudes regarding technology. In fact, individual attitudes that shape use of technology may prevail regardless of age (24). Again, supporting the notion that generations need to be redefined over time (9).

However, as we do not agree that it is well-established that age is not a relevant predictor for technology use, we did not make any revision based on this comment.

Comment: the following terms or sentences were hard to understand or suspected wrong, please consider replacing/rephrasing:

p. 3 surf tablets

p. 3 artificial body organs

p. 9 CD-recorder

p. 14 participants choose to be without products

p. 14 proc

p. 14 communicators

p. 23 more or less no support

p. 24 better-worse continuums rang

p. 24 age coding attitudes

p. 25 data mining - replace by collection of data?

Response: Changes has been made as follows: 

p. 3 tablets

p. 3 artificial body parts

p. 9 CD-player

p. 14 participants choose to be without products changed to; participants choose to use products

p. 14 proc, have been taken away

p. 14 communicators changed to; alternative communication devices

p. 23 more or less no support changed to: no support

p. 24 better-worse continuums rang changed to: better-worse continuums were evident

p. 25 data mining - replace by collection of data? As data mining is not the same as data collection – we changed the text to read: They argued for increased restrictions on how companies are using the vast amount of individual data they are collecting because current legislation, such as the General Data Protection Regulation (GDPR) within EU (that directs who is allowed to sell personal information), is not sufficiently respected. 

p. 24 age coding attitudes. As we do not quite understand the comment regarding this, we are thus unsure about what and how to revise. The concept is written in line with reference #25. 

Comment: Title

1)The title is limited to a verbatim citation of one participant. I recommend that the authors modify it to a title that readers can find what follows generally in the study. My suggestion is something like “Attitudes towards use of technologies in three generations: A qualitative study”.

2)I recommend that the authors consider some more keywords that cover the major contents of their paper. None of the three keywords that have been mentioned contain the main terms in the title too. I recommend to add some more keywords that refer to the title and main points in the paper.

Response: 1) As using a citation in the title of reports of qualitative studies is not uncommon in the scientific literature, we prefer to keep this style because it serves to elicit interest for the study. The title reflects what follows in the paper, namely that individual attitudes are present beyond generations. We have changed the title somewhat to read; “Am I representative (of my age)? No, I'm not” – Attitudes to technologies and technology development differ but unite individuals across rather than within generations

Keywords have been added

Comment: Introduction

1)Please bring a reference for “While technologies are broadly defined as goods, services, knowledge, skills etc. (p2, line 23).

2)Please bring a reference for four categorizations regarding technologies (p3, line 5) (or are they categorized by the authors? How did they categorize them?

For instance, the authors categorized eHealth solution as a welfare technology, they can be categorized as assistive technologies or as information and communication technologies as well. Or the authors categorized TV as everyday technologies, it can be categorized as information and communication technologies too. Thus, as there is a different categorisation for various types of technologies, a scientific source about how technologies have been categorized is required.

Response: 1) An reference has been added

2) Thank you for these comments. The categorization was developed for the GenerationTech project, and as it was used as a starting point during the focus groups sessions we prefer to not make any change in this respect. However, we have made some clarifications and the text now reads; 

In contrast, the present study targeted four categories including traditional and digital technologies: everyday technologies (e.g., refrigerators, kitchenware, cars, new lightbulbs, TVs), information and communication technology (e.g., smartphones, surf tablets, computers), welfare technology (e.g., safety alarms, night cameras, eHealth solutions), and medical and assistive technologies (e.g., walkers, wheelchairs and communication aids, medical devices such as artificial body parts). The two first categories are used by the general population, while the latter two are provided by the health and social care systems rather than privately bought to support active and healthy ageing. 

Comment: Method

1)The authors should bring a reason in a clearer way about why they selected such method.

2)Inclusion and exclusion criteria for composing a sample or key characteristics of the sample should be more clarified. The authors should make explicit why they compose such sample (what are the criteria for collecting such sample). This information may help other researchers if they want to replicate a similar study.

Response: 1) This text have been added on page 5, line 5: 

than would have been possible through individual interviews (22). Designing a process with age homogenous groups in the first round of sessions and age-mixed groups in the second, focus groups enabled discussions within and across generations. In this way, we were able to achieve engaging and thoughtful discussions among the participants and thereby a deeper understanding of the topics at target. 

2) Persons from the three generations able to communicate in Swedish were recruited. Some more details on the recruitment process have been added on page 5 line 13 and the text now reads:

Aiming for heterogeneity in attitudes and experiences with technologies from a broad perspective, we recruited the participants from different work and life situations through networks such as social service staff in a nearby municipality and non-governmental organizations for people with disabilities. With support from university staff, we recruited current/past students within humanities, social medicine, global health and engineering. We also approached professionals and seniors who had previously attended events organized by our research centre; we knew that they had experience from welfare technology as well as assistive technology. Homogeneity (24) in terms of age and heterogeneity (24) based on sex and educational background, work and life situations was established to generate and explore different perspectives on the study topic (25).

Comment: Findings

1)As authors consider four categories for technologies, the findings about attitudes or experiences about use should also be clarified regarding different types of technologies and their category (most verbatim citations that authors brought in this paper refer to information and communication devices).

It is an important point because technologies in different categories have specific attributes, then the attitude towards each category may be different for instance the use experience of assistive technology such as wheelchair can be different in comparison with mobile phones. Thus, it may arise different concerns.

This distinction regarding different categories of technologies should also be illustrated in Figure 1 (It is not clear the term of technologies refer to which four categories of technologies that have been mentioned by the authors). Digital technology is just one that is clear in Figure 1.

2)Preparing the main findings in a form of table regarding attitudes' similarities and differences in three generations help readers to grasp the message easier.

Response: 1) As highlighted on page 10, line 6 The findings also reflect that participants across generations preferred to discuss a broad range of digital, rather than traditional technologies. Most often the participants referred to “technologies” and the more detailed examples given by participants are already presented in the text. However, we have revised the text to use the four categories more consistently in the text, for example on page 16 line 14

A table (Table 2 has been added)

Comment: Discussion

1)In the last line of page 23, it is mentioned that the wide range of technologies considered in the present study increasingly involve digital components (please bring some examples of these technologies that involve digital components and have been considered in this paper).

2)“The presence of individual-minded rather than generational attitudes in the area of technologies and technology development is a finding rarely described in research” (P24, line 17).

This point is an important point that can be more elaborated by the authors as there are some influential models about technologies that consider individual user perspectives. The authors may emphasis which individual perspectives they discovered in their paper (please bring some examples maybe beyond the individual user factors that have already been considered in the existing models).

Response: 1) This is a relevant comment and examples have been added on page 26 line 11 e.g. smart appliances, or self-driving cars

2) We agree with the findings from Abri & Boll (2020) (reference #21) added to this paper, describing that most models neglect user perspectives (page 4, line 8 of our paper)

Comment: Conclusion

1)The authors concluded that “future technologies, as well as services and policies aiming to support the use of them should consider individual user perspectives” (p29, line 15).

As discussed in the comment above, The point regarding individual user perspective is an important point that needs to be more elaborated by bringing some examples (which aspects of individual user perspectives have not been considered in the previous model studies or in the previous qualitative studies that should be considered in the future).

In the discussion part, it is mentioned (page 24) that personality and feelings are two important things to be considered. These two factors have already been considered in some existing models of technologies.

Some examples for individual user perspective (that the authors discovered by the current study) are required here

Response: As the findings show that perspectives both differ and are similar within and across generations, we do not think it is relevant to list user perspectives. However, in line with Abri & Boll (2020) we conclude that user perspectives in terms of needs, desires, beliefs or goals are overall neglected in the existing technology models. Thus, we have added these aspects as examples of perspectives in the conclusion. As user perspectives differ, we also highlight the importance to involve users in the future technology development. As described above we have added more arguments in the introduction related to this, page 4, line 8. 

Comment: After reading the introduction, it was not entirely clear to me what is missing in the literature and how this research could answer this gap. The authors state that few studies have used qualitative methods and cross-generational perspectives but I would appreciate a more specific theoretical and methodological rationale clarifying the overall importance and novelty of this particular study.

Following up to this point, I am missing a rationale for the relevance of the overarching research topic. The authors might consider shortly discussing the role of digital technology for enhancing active and healthy aging, for example, with respect to cognitive functioning, depression or social support (e.g., Cotton et al., 2014; Czaja et al., 2017; Kamin et al., 2020).

Response: Thank you for this comment; we have revised the text on page 4, line 6 as follows: Previous studies on development of technologies (9) and attitudes toward technology use have largely utilised deductive lenses (2) such as the Technology Acceptance Model (TAM) (16, 17) or the Unified Theory of Acceptance and Use of Technology (UTAUT) (18). In addition, most of these originate from disciplines such as information system engineering, business or management science. They neglect or only implicitly consider desires, beliefs and goals of potential users (19). In contrast and in line with the underpinnings of gerontechnology, inter-disciplinary efforts integrating multiple disciplinary and user perspectives are certainly called for to address this neglect (14). In fact, few studies have adopted qualitative, user-centred research designed to create a deeper understanding of attitudes toward technologies, especially in the light of differences and similarities between and across age cohorts or generations.

Comment: The authors should explain and justify their methodological approach. There are many types of qualitative methods and I would like to know why they have focused on qualitative content analysis to address their research question.

Response: This is a relevant comment, we have now added this information on page 8, line 12: Qualitative content analysis is a flexible method appropriate when analysing data from focus groups (22), particularly relevant related to these multifaceted phenomena of technology where little knowledge exists from the generational perspective.

Comment: The authors correctly argue that technology attitudes are a complex research issue. There are many factors (e.g., cognitive, physiological, motivational, social) associated with technology use across the lifespan and therefore I was wondering whether and how this heterogeneity was reflected in the sample. If possible, it would be helpful to have more information about the sample (e.g., health status or functional limitations, previous experience with technology).

Response: Details have been added as follows: Ten lived alone and the rest were cohabiting. A majority of the participants reported being in good or very good health, and they together represented a diversity of professions.

Comment: If possible, please provide the inter-rater reliability for the categorization. This would greatly support the methodological consistency of the paper.

Response: Inter-rater reliability is not a relevant aspect of qualitative analyses. In line with the qualitative design of the present study we applied strategies to ensure a valid and trustworthy description where all authors took on different roles, as described on page 8 and 9 and supported by reference #44.

Comment: After reading the discussion, it remained unclear how the findings contribute to the existing literature in the field. What does this study contribute to the key issue of interindividual differences in (late life) technology adoption/acceptance?

Cotten, S. R., Ford, G., Ford, S., & Hale, T. M. (2014). Internet use and depression among retired older adults in the United States: a longitudinal analysis. The journals of gerontology. Series B, Psychological sciences and social sciences, 69(5), 763–771, Article Journal Article. https://doi.org/10.1093/geronb/gbu018

Czaja, S. J. (2017). The Role of Technology in Supporting Social Engagement Among Older Adults. Public Policy & Aging Report, 27(4), 145-148. https://doi.org/10.1093/ppar/prx034

Kamin, S. T., & Lang, F. R. (2020). Internet Use and Cognitive Functioning in Late Adulthood: Longitudinal Findings from the Survey of Health, Ageing and Retirement in Europe (SHARE). J Gerontol B Psychol Sci Soc Sci, 75(3), 534-539, https://doi.org/10.1093/geronb/gby123

Response: Thank you for the suggested references. In the revised manuscript we utilized two of them in the introduction: several current studies (16, 17) largely focus on various health outcomes with a particular focus on older rather than ageing populations.

---

## [Editor Report · Decision Letter 1]

7 Apr 2021

“Am I representative (of my age)? No, I'm not” – Attitudes to technologies and technology development differ but unite individuals across rather than within generations

PONE-D-20-32347R1

Dear Dr. Fristedt,

We’re pleased to inform you that your manuscript has been judged scientifically suitable for publication and will be formally accepted for publication once it meets all outstanding technical requirements.

Kind regards,

Anja K Leist, Professor Dr.

Academic Editor

PLOS ONE

Additional Editor Comments (optional):

The authors have addressed all comments sufficiently, and incorporated the relevant literature to strengthen the theoretical framing of their findings.
---

## [Editor Report · Acceptance letter]

12 Apr 2021

PONE-D-20-32347R1 

“Am I representative (of my age)? No, I'm not” – Attitudes to technologies and technology development differ but unite individuals across rather than within generations 

Dear Dr. Fristedt:

I'm pleased to inform you that your manuscript has been deemed suitable for publication in PLOS ONE. Congratulations! Your manuscript is now with our production department. 

Kind regards, 

on behalf of

Prof. Dr. Anja K Leist 

Academic Editor

PLOS ONE